# A novel pathogenesis concept of biliary atresia approached by combined molecular strategies

Wison Laochareonsuk[1,2], Komwit Surachat[2,3], Piyawan Chiengkriwate[1], Surasak Sangkhathat[1,2]*

1 Faculty of Medicine, Department of Surgery, Prince of Songkla University, Hat Yai, Songkhla, Thailand, 2 Faculty of Medicine, Translational Medicine Research Center, Prince of Songkla University, Hat Yai, Songkhla, Thailand, 3 Faculty of Medicine, Department of Biomedical Sciences and Biomedical Engineering, Prince of Songkla University, Hat Yai, Songkhla, Thailand

* surasak.sa@psu.ac.th

## Abstract

Cholestatic jaundice is one of the most common neonatal conditions. BA, a correctable cholangiopathy, presents with cholestatic jaundice within the first weeks of life. The inflammation of bile ducts leads to progressive fibrosclerosis involving biliary trees, followed by cirrhosis and liver failure. With the use of modern molecular studies, this research aimed to define a novel pathogenesis by exploring variations. We performed genetic discovery by using supervised and unsupervised approaches. Ultimately, a combination of genetic variations and survival data was analyzed to strengthen the novel concept in this study. In this study, coding regions were explored to identify rare deleterious variants within genes from the first analysis together with gene sets reported in PFIC, and diseases with hyperbilirubinemia. Our unsupervised prioritization was primarily designed to identify novel causal genes from nonsynonymous variants derived by three biostatistical algorithms: enrichment analysis, burden test, and trio study. Survival analysis was integratively evaluated with a combination of identified causal genes. The individuals with identified variants from the supervised approach were frequently related to the severity of cirrhosis and poor postoperative outcome. In the unsupervised approach, nonsynonymous variants were enriched. Cilium and muscle related pathways had a significant correlation. *CCDC8* was statistically significant gene in which six cases carried mutations identified through burden analysis. Individuals who carried variants in corresponding genes and significant pathways had significantly lower native-liver survival than individuals in whom none of these variants were identified (log-rank p value 0.016). This study explored genetic variations by multiple strategies. Different pathways of cholestatic diseases have been found to be associated with BA. Therefore, BA may be characterized as a shared sequela of many cholestatic disorders. Susceptibility in those pathways suggested an association with BA and strengthened this proposed novel hypothesis. The results emphasized the consequences of many disruptive pathophysiologies.

**Data Availability Statement:** All relevant data are within the article and its Supporting information files.

**Funding:** The study was supported by the Genomic Thailand Initiative Project through the Health System Research Institute (HSRI 63-096 and HSRI 63-080). The funders had no role in study design, data collection and analysis, decision to publish, or preparation of the manuscript.

**Competing interests:** The authors have declared that no competing interests exist.

## Introduction

Biliary atresia (BA) is an inflammatory cholangiopathy manifesting as gradual obstructive jaundice within the first two weeks of life. The primary pathology, severe periductal inflammation, leads to progressive fibrosclerosis involving both extrahepatic and intrahepatic bile ducts, followed by biliary cirrhosis, portal hypertension and fulminant liver failure [1]. Surgical bypass of the extrahepatic biliary system, hepatic portoenterostomy (HPE), is the standard treatment in patients diagnosed with BA. Without appropriate treatment, infants with BA are unlikely to survive with their native liver for longer than two years [2,3]. The highest incidence of BA is reported in Asia-Pacific regions, ranging from 1:320–1:2000 live births. It remains unclear if the pathogenesis of inflammation in BA is secondary to viral infection, an immunological process or a developmental mishap [1]. A recent genetic study by Sangkhathat et al. [4] proposed that BA might be a common pathological process of various entities, as genetic variants reported in other infantile cholestatic conditions were also detected in clinically diagnosed BA. As the molecular pathology regarding the pathophysiology of cholestatic jaundice has been more elucidated, novel diseases have been characterized, and some of those diseases may share similar clinical presentations and histopathology with BA and BA with germline variants may have different prognosis from sporadic one.

In recent decades, the spotlight on the pathogenetic theories of BA has moved to genetic susceptibility following the introduction of high-throughput genetic studies, led by genome-wide association studies (GWASs). Single-nucleotide polymorphisms (SNPs) within the locus 10q25, near the region of Adducin3 (*ADD3*), were identified as having a significant association with the occurrence of BA in various independent populations, including Han Chinese, Thai, and European populations [5–8]. Furthermore, the GWAS in Europeans localized a second SNP on locus 2p16, an intronic region of EGF, situating the Fibulin Extracellular Matrix Protein 1 (*EFEMP1*) gene as a novel candidate gene [9]. Next-generation sequencing (NGS), an evolutionary high-throughput platform, enables faster and more robust variant discovery than conventional sequencing technologies [10]. Recently, whole-exome sequencing (WES) has identified novel deleterious variants in various genes that play a role in ciliary functions (*KIF3B*, *PCNT*, and *TTC17*) in BA patients.

This study aimed to investigate a novel perspective concept in the pathogenesis of BA by exploring genetic variations and biostatistical appraisal. The molecular genetic analyses were performed using the WES platform, followed by bioinformatic analysis. Our genetic discovery algorithm consisted of supervised variant prioritization looking for previously reported genes in infantile cholestatic jaundice diseases/syndromes and an unsupervised approach which aimed to identify novel variants, followed by biostatistical analysis of all identified variants. To gain more insight into the phenotype correlation, genetic information was categorized and collectively analyzed with the clinical manifestations, the outcomes of surgical correction, and associated syndromic features.

## Materials and methods

### Patients and biomaterials

Infants with BA who underwent HPE at our institute from 2003 to 2020 were recruited into the study through informed consent from their parents. The study was approved by the Human Research Ethics Committee of the Faculty of Medicine, Prince of Songkla University (REC-61-351-10-1). The diagnosis of BA was made through intraoperative cholangiography and intraoperative findings together with additional conformation by pathological report and clinical follow-up data. Other syndromes which clinical features similar to biliary atresia e.g.,

Alagille syndrome, were excluded when there was strong diagnostic evidence such as histopathology that revealed biliary hypoplasia or paucity of bile duct. During the same operations, liver biopsy specimens were retrieved and cryo-preserved at the Human Biobank (Translational Medicine Research Center, Faculty of Medicine, Prince of Songkla University). Peripheral blood samples of the parents were obtained during the follow-up periods. Genome data from unrelated controls from the same geographic region aged between 1 and 60 years were provided by the South Thai Genome Data Bank.

Genomic DNA was extracted from the liver biopsy specimens or peripheral blood samples using a QIAGEN DNA Mini kit (QIAGEN, Hilden, Germany) and a High Pure PCR template preparation kit (Roche, Berlin, Germany). The extracted DNA was quantified for concentration and purity using a nanodrop (Thermo Scientific, Delaware, United States). The qualified DNA was subjected to archived fluorescent electrophoresis to evaluate DNA fragmentation using the Agilent Genomic DNA ScreenTape assay (Agilent Technologies, Santa Clara, California, United States).

### Clinical follow-up and data retrieval

Postoperative care after HPE consisted of systemic steroids for 2 weeks, ursodeoxycholic acid for one month, and oral antibiotics for one month unless there was an indication for extended prophylaxis. Clinical follow-up and liver function studies were scheduled every six months until five years and then annually thereafter. In addition, active surveillance of liver cirrhosis was performed once a year with ultrasonography and esophagoscopy in cases with cirrhosis and clinical signs of portal hypertension. Patients with remarkable liver deterioration were put on a waiting list for the liver transplantation program at Siriraj Hospital, Bangkok.

Clinical data were retrieved from the electronic medical records, including date of birth, date of surgery, anatomical subtype based on Kasai classification, liver function test at diagnosis (total serum bilirubin, direct bilirubin, AST, ALT, ALP, GGT and serum albumin), liver function test 1 month after surgery, operative procedure and findings, adjunctive treatment after surgery (oral antibiotics, ursodeoxycholic acid, oral-systemic steroids), last follow-up date and status. The definition of 'jaundice-clear' was decreased serum bilirubin to less than 2 mg/dL within 1–3 month after surgery, with 'jaundice-persistent' defined in those who had a persistently high level of serum bilirubin. Within the jaundice-persistent group, there were 2 subcategories: (1) jaundice improved (total bilirubin decreases more than 80% of total) and (2) jaundice not improved (total bilirubin decreases less than 20% of total).

### Whole-exome sequencing

Exome sequencing libraries were prepared, and coding regions were captured and enriched using Agilent SureSelect XT Human All Exon v6 following the manufacturer's protocol (Agilent Technologies, Santa Clara, California, United States). The library was quantified with a Qubit dsDNA High Sense Assay Kit (Invitrogen, USA). The library size was measured by an Agilent D1000 ScreenTape assay. Paired-end sequencing with a 150-bp platform was carried out on an Illumina NovaSeq-6000 (Illumina, San Diego, California, United States) at an average targeted coverage of 200x depth.

### Bioinformatic analysis

The paired-end sequence files were qualified by FastQC (version 0.11.9) and trimmed using Trimmomatic (version 0.39) with a paired-end trimming function. The optimally prepared FASTQ files were aligned with the human reference genome (version GRCh38.13) using BWA (version 0.7.17) [11]. Subsequently, the SAM (Sequence Alignment Map) was converted to a

BAM (Binary Alignment Map) and sorted by SAMtools (version 1.10) [12]. Then, the sorted BAM was regrouped and had identical sequences marked using Picard (version 2.18.26). The unduplicated BAM was adjusted for base quality score by GATK (version 4.2.0) [13] BQSR. The variants were subtracted from the exome using GATK HaplotypeCaller and stored as intermediate genotype information in GVCF (variant calling format genome VCF). When variant calling was finished for every sample, the GVCFs of those cases were merged with the disease control data using GATK CombineGCVFs and calculated for true possible variants using GATK GenotypeGVCFs. The combined variants were filtered out using a Gaussian mixture model with 99% truth sensitivity to select true variants. Finally, those variants were appended to functional annotation with Variant Effect Predictor (VEP) version 102 [14] based on dbSNP 151.

## Variant selection

All identified variants were crudely filtered using the criteria of having a reading depth more than 20x and allele frequency of individual sample more than 25%. A variant was considered to have a high or moderate impact when the minor allele frequencies were less than 0.01 in the East Asian population. A variant was included in downstream analysis when its clinical correlation was available in the Human Gene Mutation Database (University of Cardiff, United States), ClinVar [15] and OMIM (Online Mendelian Inheritance in Man) [16]. If there was no report available, the effect on protein was evaluated by Sorting Tolerant From Intolerant (SIFT) [17], Polymorphism Phenotyping data collection 2 (PolyPhen2) [18], Mutation Taster–AA change score and probability value (Mutation Taster2) [19], and Combined Annotation Dependent Depletion (CADD) [20]. The criteria for defining pathogenic consequences of a variant were prediction as damaging (SIFT), possibly deleterious or deleterious (PolyPhen2), disease-causing (Mutation Taster), and CADD Phred score over 20.

## Supervised variant prioritization

With a hypothesis that variants associated with congenital cholestatic diseases might be overlapping with variants found in those with BA phenotypes. Supervised annotation and prioritization were performed based on known candidate genes involved in cholestasis jaundice conditions: 1. progressive familial intrahepatic cholestasis (PFIC), 2. syndromes in which BA is reported as one of the phenotypes, and 3. syndromes with hyperbilirubinemia as one of the phenotypes (shown in S1–S3 Tables). The positions and nucleotide alterations of the variants were confirmed by demonstrating their differential alignment in Integrative Genomic Viewer (IGV). All confirmed variants were revalidated using PCR and Sanger sequencing.

## Unsupervised variant prioritization

Selected variants from the cases and controls were submitted for three downstream variant prioritization algorithms: case-only, case–control, and trio analysis. In the case-only study, the variants were manipulated from VCF to tab delimited format using BCFtools (version 1.10.2). The genes were enriched by the EnrichR package (version 3.0) following eight databases: Gene Ontology (GO) molecular function, GO cellular component, GO biological process, Kyoto Encyclopedia of Genes and Genomes (KEGG), Human Reactome, Panther, Human WikiPathway, and BioCarta. A pathway was considered as significant when the false discovery rate (FDR) was below the threshold of 0.05 as calculated by Benjamini and Hochberg [21].

For the case–control analysis, genotyped VCF files from both groups were converted to the PED and MAP formats using PLINK (version 1.90) [22]. The variants were qualified and filtered out high missing variants following the criterion of call rate more than 95% for each

sample using PLINK. Hardy Weinberg Equilibriums (HWE) were calculated, and the variants were considered as passed when the p value was less than 0.05. Identical-by-descent (IBD) relatedness was assessed for the relationships between the samples in both groups in which the estimated pi-hat score should be lower than 0.9. The remaining variants were evaluated for homogeneity using principal component analysis (PCA) from the IPCAPS package (version 1.1.8) [23]. Nonhomogenous samples were excluded from the study. The qualified variants were selected from the raw merged VCF depending on the variant selection criteria and reconverted to PED and MAP, which were approved to be compatible based on rvtests (rare variant test, version 20190205) software. The variants were grouped into gene sets based on refFlat gene definitions (version GRCh38.13) and burden analysis of the different conditions. The gene-set was performed using the collapsing and combine (CMC) model with the exact test. A gene was considered as significant when the FDR was below the threshold of 0.05.

Trio analysis was performed by selecting variants belonging to a mode of inheritance by fitting either autosomal dominant or recessive models. The dominant model was designed to discover denovo mutations, which are only found in probands, while the recessive model assessed variants inherited from the parents. All variants identified from the three analysis methods demonstrated differential alignments in IGV and were revalidated using PCR and Sanger sequencing.

## Statistical analysis

The statistical analysis of continuous clinical data was executed by a T-test for normally distributive data and a Mann-Whitney U test for non-normally distributive data. On the other hand, categorical data are analyzed by the chi-square ($X^2$) test. The statistical analysis is achieved on the R base package and a p-value <0.05 is taken to indicate statistical significance. The survival analysis was performed by using duration after surgery to last follow-up with native liver together with combination of clinical and genetic factors. In addition, survival analysis used a log-rank test and displayed in Kaplan-Meier curves. Cox proportional hazards model was applied to identify the correlation between survival rate and factors using survival package (version 3.2–13) and survminer package (version 0.4.9). Statistical significance was considered when P value <0.05. Boxplot and bar graph were created by using ggplot2 package (version 3.3.5) and Kaplan-Meier graph was initiated by using sur package (version 1.0.4).

## Results

### Clinical characteristics

The clinical manifestations of the 75 BA patients in the study are shown in Table 1 and Fig 1. There were 39 males (52%) and 36 females (48%). The mean age at surgery was 83 days (SD 32 days). The average total and direct bilirubin levels before surgery were 11.89 mg/dL (SD 3.84 mg/dL) and 10.38 mg/dL (SD 3.56 mg/dL), respectively. At one month following the operations, the average total and direct bilirubin levels had gradually decreased in the jaundice-improved cases to total bilirubin 4.13 mg/dL (SD 3.54 mg/dL) and direct bilirubin 3.39 mg/dL (SD 2.98 mg/dL). However, the jaundice unimproved cases had severe cholestatic jaundice with average serum total bilirubin at 17.36 mg/dL (SD 7.64 mg/dL) and direct bilirubin at 15.23 mg/dL (SD 5.81 mg/dL). In addition, serum aspartate transaminase was significantly different between the two groups (P < 0.0001), which was inferred to mean continuity of hepatitis in the jaundice not improved cases. Twenty-three of the jaundice improved patients (76.0%) survived, while 23 of the jaundice not improved (53.4%) patients survived.

**Table 1. Clinical and demographic data of the studied biliary atresia patients.**

| | Jaundice-clear (n = 32) | Jaundice-persistent (n = 43) | Total (n = 75) | *P value* |
|---|---|---|---|---|
| **Sex** | | | | |
| **Male** | 17 (47.2%) | 19 (38.5%) | 39 (52%) | 0.443[a] |
| **Female** | 15 (52.8%) | 24 (61.5%) | 36 (48%) | |
| **Age at surgery** (Mean ± S.D.) | 86 ± 38 | 80 ± 25 | 83 ± 32 | 0.043[b] |
| **Serum bilirubin before surgery** | | | | |
| **Total Bilirubin** (mg/dL) (Mean ± S.D.) | 12.13 ± 3.81 | 11.57 ± 3.93 | 11.89 ± 3.84 | 0.301[b] |
| **Directed Bilirubin** (mg/dL) (Mean ± SD.) | 10.75 ± 3.44 | 9.89 ± 3.63 | 10.38 ± 3.56 | 0.535[b] |
| **Serum bilirubin 1 month after surgery** | | | | |
| **Total Bilirubin** (mg/dL) (Mean ± S.D.) | 4.13 ± 3.54 | 17.36 ± 7.64 | 9.78 ± 8.65 | <0.001[b] |
| **Directed Bilirubin** (mg/dL) (Mean ± SD.) | 3.39 ± 2.98 | 15.23 ± 5.81 | 8.43 ± 7.34 | <0.001[b] |
| **Liver function test 1 month after surgery** | | | | |
| **AST** (Unit/L) (Mean ± S.D.) | 146.19 ± 71.81 | 327.59 ± 165.59 | 223.63 ± 142.12 | <0.001[b] |
| **ALT** (Unit/L) (Mean ± S.D.) | 138.95 ± 126.76 | 202.84 ± 147.08 | 169.55 ± 151.15 | 0.052[b] |
| **ALP** (Unit/L) (Mean ± S.D.) | 594.05 ± 305.42 | 683.66 ± 533.19 | 642.42 ± 435.73 | 0.361[b] |
| **Anatomical subtype** | | | | |
| Type I | 1 (3.13%) | 2 (4.65%) | 3 (4.00%) | |
| Type II | 2 (6.25%) | 2 (4.65%) | 4 (5.33%) | |
| Type III | 29 (90.62%) | 39 (90.69%) | 68 (90.67%) | 0.907[a] |
| **Alive** | 23 (50.0%) | 23 (50.0%) | 46 (61.3%) | 0.106[a] |
| **Died** | 9 (31.0%) | 20 (69.0%) | 29 (38.7%) | |

SD, standard deviation; AST, serum aspartate transaminase; ALT, serum alanine aminotransferase; ALP, serum alkaline phosphatase.

[a] Chi-square test or Fisher exact test.

[b] T-test or Wilcoxon signed-rank test.

## Supervised variant prioritization

The variants in the related cholestatic jaundice genes were filtered and prioritized from a combined genotype VCF by selecting the Entrez gene symbol. The twenty-six variants identified in 22 of the BA cases were *ABCB4*, *ABCC2*, *ATP8B1*, *JAG1*, *KCNH1*, *KMT2D*, *MYO5B*, *SLCO1B1*, *TJP2*, and *UGT1A1* (shown in Table 2). All prioritized variants were consistent with the criteria of moderate or high impact variants, no reported benign or likely benign in the ClinVar database, low allele frequency in the EAS population, or predicted deleterious function by SIFT, PolyPhen2, MutationTaster, or CADD. *KMT2D* was the most frequently prioritized variant, identified in 6 cases. Half of the cases with variants in this group had decompensated cirrhosis and sequelae of portal hypertension, e.g., hypersplenism, esophageal varices, or ascites. Five cases of BA eventually underwent liver transplantation after developing liver failure (shown in Table 3). All identified variants were validated by visualization alignment on IGV software and confirmed by Sanger sequencing.

## Unsupervised variant prioritization

**Case-only analysis (enrichment analysis).** In the case-only analysis, 254,990 variants from the cases recorded in VCF format were transformed to a tab-delimited format and imported to the R program by which the Entrez gene symbol was appended. Enrichment analysis was performed for 106 coding genes using the EnrichR package together with 8 pathway databases. The remaining three databases with significant pathway enrichment were GO

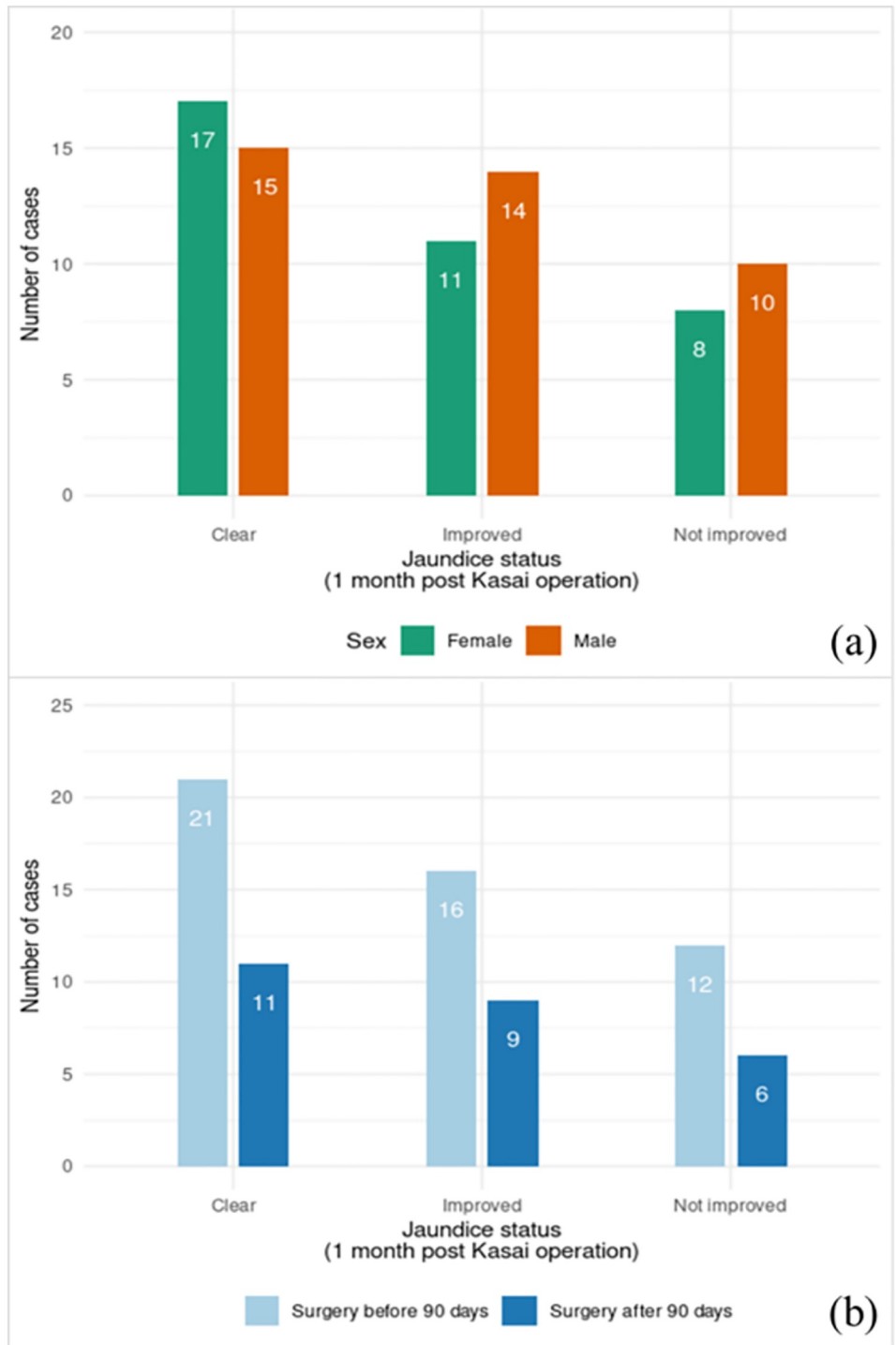

**Fig 1. Total BA Cases Stratified by Jaundice Status Together with Sex (A) and Day of Surgery (B).**

biological process, WikiPathway, and Reactome. There were 18 pathways enriched from the GO biological process database: skin development (GO:0043588), cilium assembly (GO:0060271), cilium organization (GO:0044782), actin-myosin filament sliding (GO:0033275), muscle filament sliding (GO:0030049), sensory perception (GO:0007600),

**Table 2. Rare variants identified by whole-exome studies from related cholestatic jaundice genes.**

| Case | Chr: position | SNP ID | Gene | Variant Effect | Nucleotide change (Zygosity) | Amino acid change | MAF | SIFT | PolyPhen2 | Mutation Taster | CADD |
|---|---|---|---|---|---|---|---|---|---|---|---|
| B007 | chr20:10656450 | rs876660980 | JAG1 | High | c.703C>T (het) | p.Arg235Ter | | | | | |
| B019 | chr18:57652525 | rs754583409 | ATP8B1 | Moderate | c.3220G>A (het) | p.Val1074Ile | 0.00 | 0.12 (T) | 0.906 (P) | 1 (D) | 23.4 |
| B021 | chr12:21205930 | | SLCO1B1 | Moderate | c.1394G>A (het) | p.Cys465Tyr | | 0 (D) | 1 (P) | 0.999992 (D) | |
| B022 | chr12:49040724 | rs201581582 | KMT2D | Moderate | c.7046C>T (het) | p.Pro2349Leu | 0.01 | 0.01 (D) | 0.006 (B) | | 22.7 |
| B025 | chr12:21224811 | rs377350683 | SLCO1B1 | Moderate | c.1837T>C (het) | p.Cys613Arg | 0.00 | 0 (D) | 0.999 (P) | 1 (D) | 25.7 |
| B031 | chr18:49837733 | rs756813138 | MYO5B | Moderate | c.4922G>A (het) | p.Arg1641His | 0.00 | 0 (D) | 0.972 (P) | 0.999983 (D) | 27.5 |
| B031 | chr10:99819250 | | ABCC2 | High | c.2601_2602insTA (het) | p. Pro868TyrfsTer21 | | | | | |
| B032 | chr12:49040712 | rs749441161 | KMT2D | Moderate | c.7058C>T (het) | p.Pro2353Leu | 0.00 | 0 (D) | 0 (B) | 0.518213 (P) | 23.3 |
| B040 | chr20:10641521 | | JAG1 | Moderate | c.2855C>A (het) | p.Ser952Tyr | | 0 (D) | 0.251 (B) | 0.99765 (D) | |
| B044 | chr12:49040922 | | KMT2D | Moderate | c.6848A>T (het) | p.Lys2283Met | | 0 (D) | 0.753 (P) | 0.997967 (D) | |
| B048 | chr2:233768226 | rs34946978 | UGT1A1 | Moderate | c.1091C>T (het) | p.Pro364Leu | 0.01 | 0 (D) | 0.999 (P) | 1 (D) | 24 |
| B052 | chr18:50040256 | rs117920737 | MYO5B | Moderate | c.197A>C (het) | p.Asp66Ala | 0.00 | 0 (D) | 0.019 (B) | 0.999999 (D) | 24.1 |
| B052 | chr2:233768226 | rs34946978 | UGT1A1 | Moderate | c.1091C>T (het) | p.Pro364Leu | 0.01 | 0 (D) | 0.999 (P) | 1 (D) | 24 |
| B053 | chr9:69249400 | rs777739913 | TJP2 | Moderate | c.2999C>T (het) | p.Pro1000Leu | 0.00 | 0.01 (D) | 0.813 (P) | 0.998527 (D) | |
| B067 | chr20:10641492 | rs527420845 | JAG1 | Moderate | c.2884A>G (het) | p.Thr962Ala | 0.00 | 0 (D) | 0.881 (P) | 1 (D) | 25.9 |
| B080 | chr12:49026295 | rs3782356 | KMT2D | Moderate | c.15671G>A (het) | p.Arg5224His | 0.01 | 0 (D) | 0.096 (B) | | 24 |
| B085 | chr1:210683976 | rs772578710 | KCNH1 | Moderate | c.2194C>T (het) | p.Arg732Trp | 0.00 | 0.03 (D) | 0.489 (P) | 0.924834 (D) | 27.3 |
| B087 | chr7:87422149 | rs540946679 | ABCB4 | Moderate | c.2288T>C (het) | p.Ile763Thr | 0.00 | 0.03 (D) | 0.127 (B) | 0.950034 (D) | 23.4 |
| B089 | chr12:49034450 | rs535351117 | KMT2D | Moderate | c.10467G>T (het) | p.Gln3489His | 0.00 | 0.25 (T) | 0.956 (P) | 0.958884 (D) | 20.7 |
| B093 | chr12:20901442 | | SLCO1B3 | Moderate | c.1756A>G (het) | p.Arg586Gly | | | | | |
| B097 | chr2:233772309 | rs114982090 | UGT1A1 | Moderate | c.1352C>T (het) | p.Pro451Leu | 0.01 | 0 (D) | 1 (P) | 1 (D) | 25.9 |
| B099 | chr12:20875331 | | SLCO1B3 | Moderate | c.740C>T (het) | p.Pro247Leu | | | | 1 (D) | |
| B105 | chr12:20875331 | | SLCO1B3 | Moderate | c.740C>T (het) | p.Pro247Leu | | | | 1 (D) | |
| B105 | chr12:49034450 | rs535351117 | KMT2D | Moderate | c.10467G>T (het) | p.Gln3489His | 0.00 | 0.25 (T) | 0.956 (P) | 0.958884 (D) | 20.7 |
| B107 | chr18:49847198 | | MYO5B | Moderate | c.4407G>A (het) | p.Met1469Ile | | 0 (D) | 0.418 (B) | 0.999996 (D) | |
| B107 | chr7:87406517 | rs764071235 | ABCB4 | High | c.3280-2A>G (het) | | 0.00 | | | | |

Chr, chromosome; SNP, single nucleotide polymorphism; HET, heterozygous; HOM, homozygous; MAF, minor allele frequency in East Asian populations; SIFT, scale-invariant feature transform (T, tolerated; D, damaging); PolyPhen, Polymorphism Phenotyping data collection (B, benign; P, possibly deleterious; D, deleterious); Mutation Taster, Mutation Taster–AA change score and probability value (D, disease-causing; P, polymorphism); CADD, Combined Annotation Dependent Depletion recorded in Phred score.

sensory perception of light stimulus (GO:0050953), inner ear morphogenesis (GO:0042472), retinal cone cell development (GO:0046549), retinal cone cell differentiation (GO:0042670), complement activation, alternative pathway (GO:0006957), parallel actin filament bundle assembly (GO:0030046), positive regulation of cardiac epithelial to mesenchymal transition (GO:0062043), peptide cross-linking (GO:0018149), and organelle assembly (GO:0070925), as shown in S4 Table.

With WikiPathway, four pathways, the striated muscle contraction pathway (WP383), Joubert syndrome (WP4656), ciliopathies (WP4803), and arrhythmogenic right ventricular

**Table 3. Clinical features of studied ba patients with associated possibly pathogenic variants identified from related cholestatic jaundice genes.**

| Case | Sex | Identified genes | Initial bilirubin (DB/TB) (mg/dL) | Age at surgery (days) | Jaundice clearance | Last follow-up age (status) | Last follow-up bilirubin (DB/TB) (mg/dL) | Follow-up remarks |
|---|---|---|---|---|---|---|---|---|
| B007 | F | JAG1 | 9.12/9.12 | 77 | Persistent | 17 years old (Alive) | 0.04/0.27 | Post-liver transplantation |
| B019 | M | ATP8B1 | 10.53/12.07 | 55 | Persistent | 2 years old (Death) | 5.09/6.81 | Cirrhosis with liver failure, hepatic encephalopathy |
| B021 | M | SLCO1B1 | 13.62/13.8 | 141 | Clear | 15 years old (Alive) | 1.93/2.46 | Cirrhosis with portal hypertension and esophageal varices |
| B022 | M | KMT2D | 7.17/9.3 | 60 | Clear | 17 years old (Alive) | 0.72/1.14 | Post-liver transplantation |
| B025 | F | SLCO1B1 | 14.75/18.82 | 106 | Persistent | 2 years old (Death) | 17.20/21.31 | Cirrhosis with liver failure and portal hypertension |
| B031 | F | MYO5B, ABCC2 | 19.92/19.93 | 160 | Persistent | 11 months (Death) | 7.15/9.79 | Cirrhosis with liver failure |
| B032 | M | KMT2D | 8.91/8.69 | 27 | Clear | 1 year old (Loss-to-follow-up) | 2.18/4.04 | Cirrhosis with liver failure and portal hypertension |
| B040 | F | JAG1 | 7.24/9.5 | 62 | Persistent | 4 months (Death) | 18.89/16.58 | Cirrhosis with liver failure |
| B044 | F | KMT2D | 8.53/11.04 | 55 | Clear | 15 years old (Alive) | 0.14/0.27 | Cirrhosis |
| B048 | M | UGT1A1 | 12.47/12.75 | 72 | Persistent | 12 years old (Alive) | 0.43/0.93 | Post liver transplantation |
| B052 | F | MYO5B, UGT1A1 | 11.25/14.48 | 114 | Clear | 12 years old (Alive) | 8.54/10.46 | Cirrhosis with hypersplenism, registered for liver transplantation |
| B053 | F | TJP2 | 9.35/11.46 | 147 | Clear | 12 years old (Alive) | 0.15/0.33 | Cirrhosis |
| B067 | F | JAG1 | 23.66/27.48 | 61 | Persistent | 4 months (Death) | 11.1/11.45 | Cirrhosis with liver failure |
| B080 | M | KMT2D | 12.37/15.76 | 79 | Persistent | 2 years old (Death) | 8.98/10.25 | Cirrhosis with liver failure |
| B085 | M | KCNH1 | 12.99/13.01 | 112 | Persistent | 8 years old (Alive) | 0.13/0.42 | Post liver transplantation |
| B087 | M | ABCB4 | 8.62/9.95 | 81 | Clear | 7 years old (Alive) | 0.14/0.30 | - |
| B089 | F | KMT2D | 9.78/11.97 | 84 | Clear | 7 months (Death) | 17.53/20.04 | Cirrhosis with liver failure |
| B093 | M | SLCO1B3 | 10.41/13.07 | 195 | Clear | 10 years old (Alive) | 0.32/0.76 | Cirrhosis with portal hypertension |
| B097 | M | UGT1A1 | 17.05/19.02 | 54 | Clear | 11 years old (Alive) | 0.22/0.57 | Post liver transplantation |
| B099 | M | SLCO1B3 | 9.21/10.83 | 76 | Clear | 5 years old (Alive) | 0.11/0.31 | - |
| B105 | M | SLCO1B3, KMT2D | 8.59/9.97 | 51 | Clear | 5 years old (Loss-to-follow-up) | 0.09/0.21 | - |
| B107 | M | MYO5B, ABCB4 | 7.00/8.62 | 64 | Persistent | 2 years old (Alive) | 0.13/1.14 | Cirrhosis with portal hypertension |

F, female; M, male; DB, direct bilirubin; TB, total bilirubin.

cardiomyopathy (WP2118), were statistically and significantly enriched from the genes identified in the BA cases, as shown in S5 Table.

For Reactome, Striated Muscle Contraction Homo sapiens (R-HSA-390522) and Muscle Contraction Homo sapiens (R-HSA-397014) were considered to have a statistically significant association with the variants from the cases. The enriched genes from all databases are displayed in Fig 2 and shown in S6 Table.

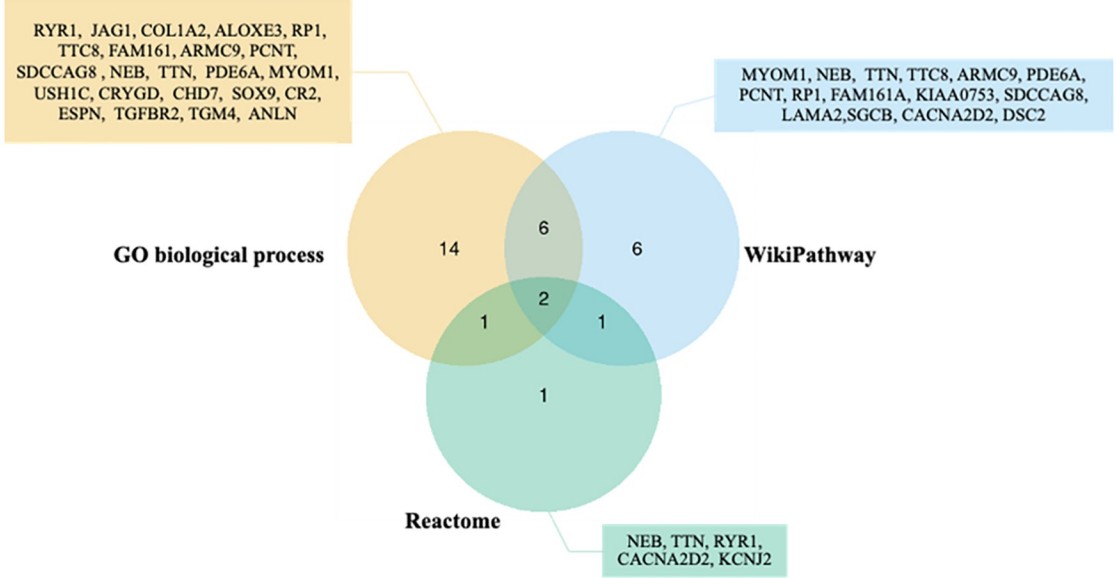

**Fig 2. Venn diagram of identified genes in 3 different pathway databases.**

**Case–control analysis (Burden analysis).** Seventy-five BA cases and 358 controls were included in the case–control analysis. The 254,990 filtered variants in the coding regions were stored in a single aggregated VCF file. The VCF was converted into PED and MAP files by PLINK software. The phenotypes and sexes of both groups were corrected using R software. The variants were assessed for SNP call rate and IBD relatedness by PLINK software. Sixteen disease control samples had SNP call rates lower than 95% and were excluded from the study; however, the estimated pi-hat score was lower than 0.9 for every sample. The allelotypes of each variant were calculated for Hardy-Weinberg equilibrium, and 15,002 variants were not in equilibrium. A PCA of those samples was created using IPCAP software (S1 Fig). Three BA case outliers were excluded from the analysis.

After filtering, 239,988 variants of 72 BA cases and 342 disease controls remained for burden analysis, which was performed based on gene region (refFlat version hg38) using rvtests software with the CMC exact test. Coiled-coil domains containing 8 (*CCDC8*) were significantly different among the cases and disease controls with FDR 0.005 (S6 Table).

**Trio analysis.** Seven BA families were included in the trio analysis based on the mode of inheritance. Variant prioritization was achieved following a dominant and recessive model. In the dominant model, two different denovo variants in *KMT2D* were detected in unrelated families. However, those variants were later reviewed during the first approach. The result in this section was confirmed as potentially causal variants. There were no nonsynonymous variants inherited from any parents in the recessive model.

## Survival analysis of clinical and genomic data

Survival analysis showed that the 5-year median native liver survival for all 75 patients was 1.8 years. Survival in patients who underwent HPE before 90 days of life was significantly better than in those who underwent surgery later than this cutoff age (log-rank p value 0.017), as shown in Fig 3.

After identifying the associated genes by unsupervised and supervised variant prioritization, the cases who carried the variants in any PFIC gene set (as shown in S1 Table), cilium-

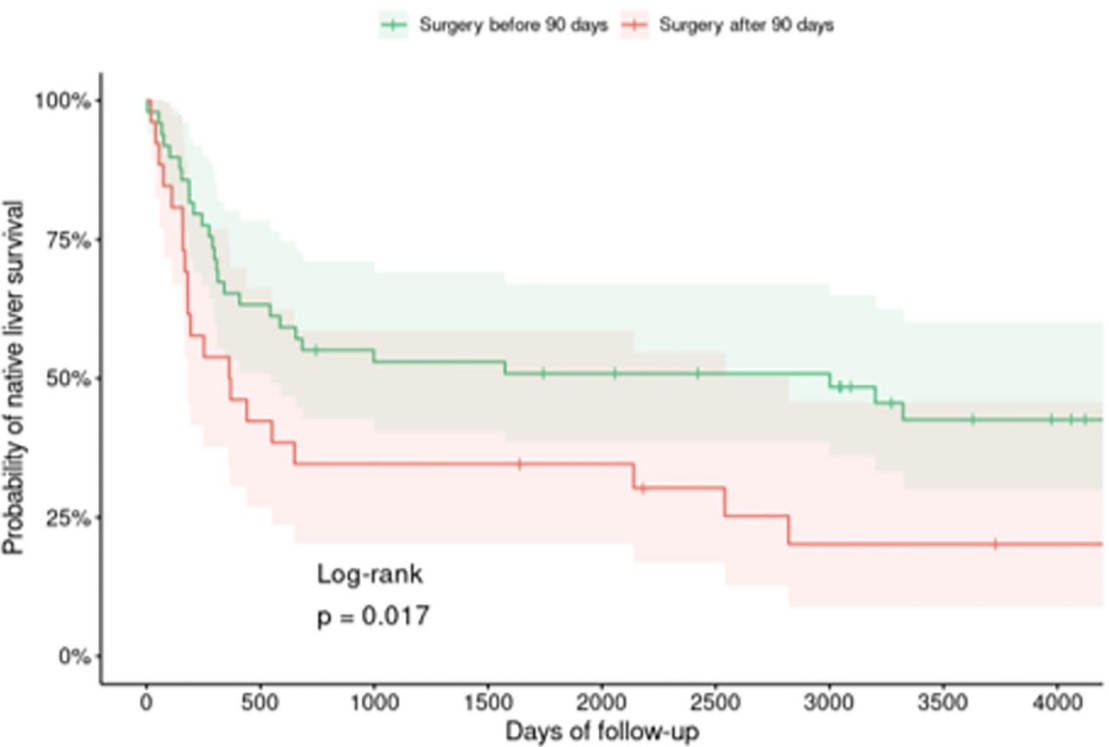

**Fig 3. Kaplan–Meier survival graph of the cases who underwent an operation before or after 90 days.**

associated genes (GO:0060271, GO:0044782, WP4803), muscle contraction-associated genes (GO:0033275, GO:0030049, WP383, R-HSA-397014), *EFEMP1*, *Sox9*, or *CCDC8* had significantly lower 5-year native liver survival (25.0%) than those without any variants (50.0%), log-rank p value 0.016 (Fig 4).

A Cox proportional hazards model was used to assess differences between survival rates and the identified variables. When assessed with a multivariate model, patients who underwent surgery before 90 days of life and who carried variants in any of the gene sets independently had a significantly increased risk of death, with hazard ratios of 1.8 (1.02–3.30) and 2.1 (1.19–3.80), respectively (Fig 5).

## Discussion

Genetic predisposition has been a focus of studies on the pathogenesis of several congenital diseases after high-throughput genome technology became available. GWAS was the first high-throughput technology to explore the genetic associations of BA in various populations. Validated in Thai and other ethnic groups, polymorphisms in the *ADD3* and Glypican 1 (*GPC1*) regions have been reported to be influentially associated with the disease and have significant gene–gene interactions [5–8]. In addition, a single GWAS in Caucasians identified a genetic association with SNPs in the *EFEMP1* region [9]. However, follow-on fine mapping studies failed to find a single gene responsible for the BA phenotype. Studies have found that in BA cases, variants known to be associated with other cholestatic diseases could be annotated, and it has been proposed that the inflammatory cholangiopathy found in BA might be a common pathological pathway of multiple disease entities [4].

In this study, nonsynonymous variants were identified through WES in 75 BA cases by two strategic approaches: supervised and unsupervised variant prioritizations. Initially, the coding

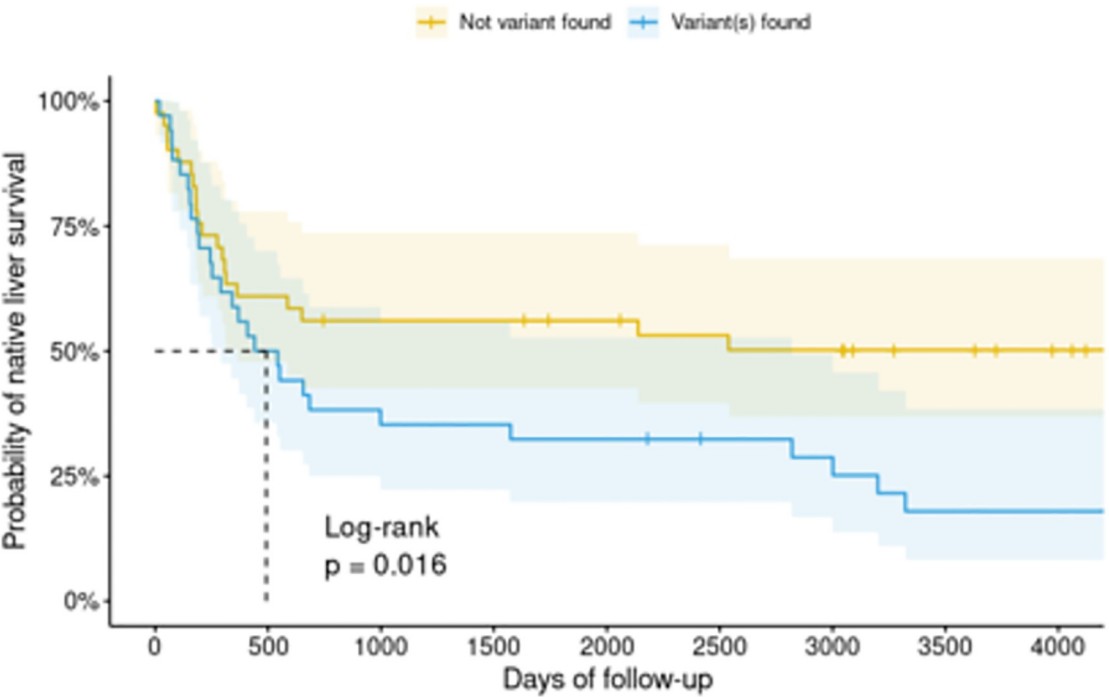

**Fig 4. Kaplan–Meier survival graph of the cases who carried the variants in associated genes.**

sequence variants were filtered according to criteria concerning the depth of coverage, allele depth, impact, minor allele frequency, and predicted deleterious function. For the supervised approach, the study has expanded the discovery of associated variants by prioritizing nonsynonymous variants located in reported gene sets of PFIC, genetic syndromes with BA as a phenotype, and diseases with hyperbilirubinemia features. The cases with candidate genes tended to have mimic phenotypes of BA and poor clinical outcomes as determined by poor postoperative jaundice clearance and early development of biliary cirrhosis.

The unsupervised approach was primarily designed to identify novel causal genes that could help to explain the pathogenesis of BA by exploring rare and deleterious variants by biostatistical analysis. In this approach, the deleterious variants were derived by three biostatistical methods: enrichment analysis, burden testing, and trio studies. The 31 identified coding genes were statistically enriched from 254,990 nonsynonymous variants of the cases.

Cilium and muscle contraction-related functional pathways were surprisingly identified to have the most significant correlation among the three databases. Seven ciliary-associated genes were extracted from this analysis: *RP1*, *TTC8*, *FAM161A*, *DNAH5*, *ARMC9*, *PCNT*, and *SDCCAG8*. Cholangiocyte ciliopathy has been suspected to be involved in the pathogenesis of BA since the genes identified by WES (*KIF3B*, *PCNT*, and *TTC17*) are involved in ciliary biological processes [24]. Ciliogenesis is required for the optimal development and regular function of cholangiocytes [25]. Cilia are located on the apical membrane of the inner lumen of the biliary epithelium and have a physiologic function as promotors of bile flow and sensors of bile components and osmolarity [26]. Deleterious mutations of ciliary genes have been reported to disrupt ciliary structure and function, leading to ciliopathy [27]. Stagnation of bile flow may lead to destructive inflammation and fibrosclerosis. In addition to the group of ciliopathy genes, the Hedgehog (Hh) signaling pathway plays an essential regulatory role in hepatobiliary embryogenesis. Hh signaling is transduced by binding of the Hh ligand on primary

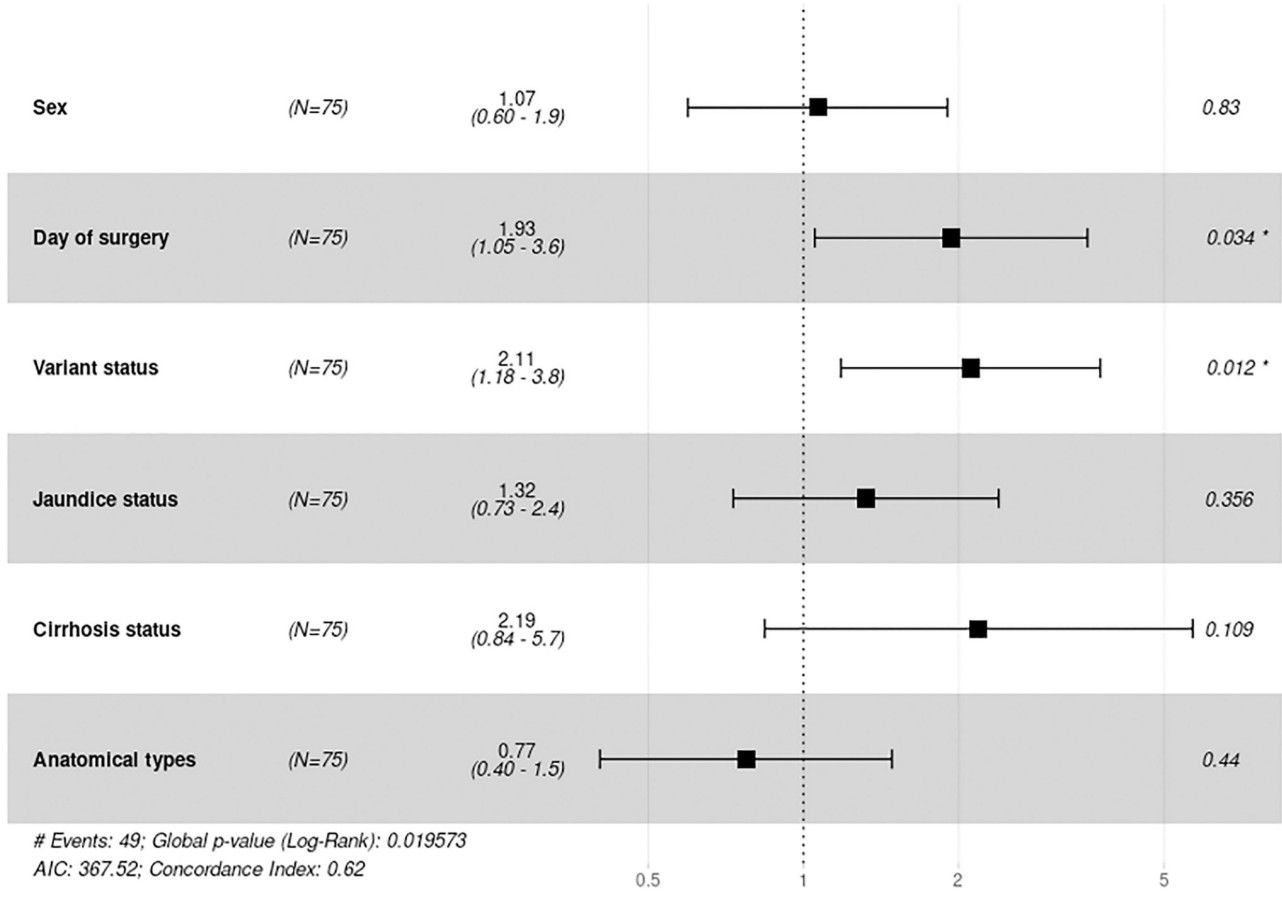

**Fig 5. Forest plot of the multivariate cox proportional hazards model of survival data and genetic variations.**

cilia, by which sequential phosphorylation of kinases is suppressed. Because of the inhibition of downstream pathways, the transcription factor translocates to the nucleus and promotes gene transcription. The Hh pathway is physiologically inactive due to the absence of Hh ligand expression in healthy individuals [28]. Hh signaling is triggered during liver damage, leading to hepatocyte regeneration [29]. *Sox9*, identified in our study, is one of the downstream molecules of the Hh signaling pathway, which is essential for hepatobiliary embryogenesis, microtubule remodeling, and biliary tubulogenesis [30]. In particular, *Sox9* may have an association with BA according to the shared pathogenesis of biliary development.

Five muscle contraction-related genes (*RYR1*, *MYOM1*, *TNNT1*, *NEB*, and *TTN*) were found to be statistically and significantly enriched. A burden test was performed for the second analysis to explore deleterious and rare causative genes between the cases and controls. *CCDC8* was the only statistically significant gene in which six cases carried heterozygous missense mutations. *CCDC8* encodes microtubule regulatory proteins and stabilized proteins that cause 3-M syndrome [31]. In addition, a cholangiocyte primary cilium consists of a microtubule-based axoneme and microtubule-organizing center, which influences normal biliary physiology. Trio-based analysis was performed in seven BA families. Two de novo missense variants in *KMT2D* were identified in two unrelated BA families. The results of this section are the putative loss-of-function of *KMT2D*, which was discovered in the supervised prioritization approach.

A multivariate Cox proportional hazards model was applied to adjust the time-to-event covariation of clinical and genetic factors. However, anatomic features of biliary atresia were one of statistical interference of the model according to previous studies [32,33]. After adjusted analysis, the analysis showed an independently increasing hazard risk in BA cases whose age at surgery was more than 90 days and those with identified variant status. To correlate with survival data, the individuals who carried nonsynonymous variants in *CCDC8*, *Sox9*, and the gene sets of *PFIC*, ciliary, and muscle contraction biological pathways had significantly lower native liver survival compared to those who did not have any variants in those genes.

Genetic susceptibility has been one of the most studied potential pathogeneses of BA. Many studies have successfully identified several different genes involved in biliary development and function in BA. Moreover, BA has been proposed as a complementary feature in various syndromes that frequently have heterogeneity in clinical manifestation. We suggest that BA could be a shared sequela of a number of obstructive biliary diseases. Biliary tract malformation, insufficiency of physiologic bile clearance, and alteration of bile components are all possibly involved in the initial pathophysiology of biliary obstruction leading to accumulation of bile components in the biliary system and activation of inflammatory cascades. Periduct inflammation progressively damages cholangiocytes and the surrounding mesenchyme and transforms the typical duct structure to irreversible fibrosclerosis found in both the operative findings and histopathology of BA.

In conclusion, coding sequences of the cases were explored in this study to discover rare and nonsynonymous variants in the gene sets described in biliary obstructive diseases or syndromes. Together with unsupervised variant prioritization, cilia and biological muscle pathways were statistically computed to have significant enrichment. A recent study reported that nonsynonymous variants in primary cilia genes were involved in the pathogenesis of BA. The burden test, a genetic association study of rare variants, contributed to the discovery of novel rare associated genes in BA. *CCDC8*, a microtubule coding gene, was the recently reported gene with a significant association with BA as identified by various computational algorithms. Ultimately, the combination of susceptibility identified by many approaches was correlated with poor native liver survival and independently associated with an increased hazard risk of survival with age at surgery. The results of survival analysis emphasized the hypothesis concerning the associations of identified variants with BA, which may be a consequence of various form of congenital cholestatic diseases.

## Supporting information

**S1 Fig. Principal component analysis of snps in the cases and disease controls.**
(TIF)

**S1 Table. Types of progressive familial intrahepatic cholestasis and reported candidate genes.**
(DOCX)

**S2 Table. Syndromes in which BA is one of the phenotypes.**
(DOCX)

**S3 Table. Syndromes with hyperbilirubinemia.**
(DOCX)

**S4 Table. Significant enrichment pathways from the GO biological process database.**
(DOCX)

**S5 Table. Significant enrichment pathways from human WikiPathway database.**
(DOCX)

**S6 Table. Significant enrichment pathways from human Reactome database.**
(DOCX)

**S7 Table. Results of burden analysis computed by rvtests software.**
(DOCX)

## Acknowledgments

Dave Patterson edited the English in the manuscript.

## Author Contributions

**Conceptualization:** Wison Laochareonsuk, Surasak Sangkhathat.

**Data curation:** Wison Laochareonsuk, Piyawan Chiengkriwate.

**Formal analysis:** Wison Laochareonsuk, Surasak Sangkhathat.

**Investigation:** Wison Laochareonsuk.

**Methodology:** Wison Laochareonsuk, Surasak Sangkhathat.

**Project administration:** Surasak Sangkhathat.

**Software:** Komwit Surachat.

**Supervision:** Surasak Sangkhathat.

**Validation:** Wison Laochareonsuk.

**Visualization:** Wison Laochareonsuk, Komwit Surachat.

**Writing – original draft:** Wison Laochareonsuk.

**Writing – review & editing:** Surasak Sangkhathat.

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
