## [Decision Letter · Decision Letter 0]

5 Oct 2022

PONE-D-22-12933A novel pathogenesis concept of biliary atresia approached by combined molecular strategiesPLOS ONE Dear Dr. Sangkhathat,

Thank you for submitting your manuscript to PLOS ONE. After careful consideration, we feel that it has merit but does not fully meet PLOS ONE’s publication criteria as it currently stands. Therefore, we invite you to submit a revised version of the manuscript that addresses the points raised during the review process.

I would like to sincerely apologise for the delay you have incurred with your submission. It has been exceptionally difficult to secure reviewers to evaluate your study. We have now received three completed reviews; the comments are available below. Although two reviewers are very positive about your study, another reviewer has raised significant scientific concerns about the study that need to be addressed in a revision

Please revise the manuscript to address all the reviewer's comments in a point-by-point response in order to ensure it is meeting the journal's publication criteria. Please note that the revised manuscript will need to undergo further review, we thus cannot at this point anticipate the outcome of the evaluation process.

We look forward to receiving your revised manuscript.

Kind regards,

Miquel Vall-llosera Camps

Senior Editor

PLOS ONE

Journal Requirements:

“The study received a grant-in-aid from the Genomic Thailand Initiative Project through the Health System Research Institute (HSRI 63-096 and HSRI 63-080). Dave Patterson edited the English in the manuscript.”

‘The funders had no role in study design, data collection and analysis, decision to publish, or preparation of the manuscript.”

Reviewers' comments:

Reviewer's Responses to Questions

**Comments to the Author**

1. Is the manuscript technically sound, and do the data support the conclusions?

Reviewer #1: Yes

Reviewer #2: Yes

Reviewer #3: Yes

2. Has the statistical analysis been performed appropriately and rigorously? 

Reviewer #1: Yes

Reviewer #2: Yes

Reviewer #3: Yes

3. Have the authors made all data underlying the findings in their manuscript fully available?

Reviewer #1: Yes

Reviewer #2: Yes

Reviewer #3: Yes

4. Is the manuscript presented in an intelligible fashion and written in standard English?

Reviewer #1: Yes

Reviewer #2: Yes

Reviewer #3: Yes

5. Review Comments to the Author

Reviewer #1: This is a nice study describing novel pathogenetic concept of biliary atresia by combined molecular strategies

The method section is ok

Results are also ok

Discussion is adequate

I have no further suggestions

Reviewer #2: As shown in Table 2, of the 75 enrolled cases, twenty-two cases with genetic variants which were related to cholestatic disorders were identified.

It may be an important finding to identify the variants in the related cholestatic jaundice genes in biliary atresia frequently. On the other hand, it is important to show the diagnosis of biliary atresia is correctly made in the subject cases. The authors should show that the genetic abnormality overlaps in the biliary atresia cases after other cholestatic diseases were correctly differentiated.

Based on this point of view, the authors should present the anatomical pattern of biliary atresia studied in these cases, referring to the following article. (1: Anatomical patterns of biliary atresia including hepatic radicles at the porta hepatis influence short- and long-term prognoses. Pancreat Sci. 2021 Nov;28(11):931-941., 2: The anatomic pattern of biliary atresia identified at the time of Kasai hepatoportoenterostomy and early postoperative clearance of

Jaundice are significant predictors of transplant-free survival. Ann Surg. 2011 Oct; 254(4): 577-585.)

In addition, the authors can demonstrate that you have correctly diagnosed cholestatic diseases when the authors show the signs of Alagille syndrome and the findings of liver biopsy.

In addition, the authors conducted a multivariate analysis, including the day of surgery and variant status. Since it has been shown that the anatomical pattern of biliary atresia also has a significant impact on prognosis, we believe that a multivariate analysis, including these factors, would provide important insights.

Reviewer #3: NONE

6. PLOS authors have the option to publish the peer review history of their article (what does this mean?). If published, this will include your full peer review and any attached files.

Reviewer #1: No

Reviewer #2: No

Reviewer #3: **Yes: **Sujit Mohanty

---

## [Author Response · Author response to Decision Letter 0]

13 Oct 2022

PLOS ONE

Dear Editor-in-chief,

 The authors thank all reviewers for their valuable suggestion for the manuscript PONE-D-22-12933 entitled “A novel pathogenesis concept of biliary atresia approached by combined molecular strategies”. The authors have carefully read the suggestion and improved the manuscript accordingly. 

Our response to the comments of the reviewer:

All authors have read and approved of these revisions, which we hope they would meet with your approval following answer statements in below:

Reviewer #2

1. As shown in Table 2, of the 75 enrolled cases, twenty-two cases with genetic variants which were related to cholestatic disorders were identified. It may be an important finding to identify the variants in the related cholestatic jaundice genes in biliary atresia frequently. On the other hand, it is important to show the diagnosis of biliary atresia is correctly made in the subject cases. The authors should show that the genetic abnormality overlaps in the biliary atresia cases after other cholestatic diseases were correctly differentiated.

Answer: The diagnosis of biliary atresia cases in this study was deepened on intraoperative cholangiography and intraoperative findings together with additional conformation by pathological report and clinical follow up. (Page5 Line7)

2. Based on this point of view, the authors should present the anatomical pattern of biliary atresia studied in these cases, referring to the following article. (1: Anatomical patterns of biliary atresia including hepatic radicles at the porta hepatis influence short- and long-term prognoses. Pancreat Sci. 2021 Nov;28(11):931-941., 2: The anatomic pattern of biliary atresia identified at the time of Kasai hepatoportoenterostomy, and early postoperative clearance of Jaundice are significant predictors of transplant-free survival. Ann Surg. 2011 Oct; 254(4): 577-585.)

Answer: We already added anatomical type of biliary atresia identified intraoperatively in our manuscript. The majority of anatomy was type III which had fibrosis at porta hepatis (68 cases, 90.67%). After we included the anatomical factors into analysis of multivariate cox proportional hazards model, the results revealed independent correlation increasing hazard risk in BA cases whose age at surgery was more than 90 days and those with identified variant status. In addition, we also cited two suggestion articles in the materials and method section (Page6 Line10). With this addition, the result was reported in Table1.

3. In addition, the authors can demonstrate that you have correctly diagnosed cholestatic diseases when the authors show the signs of Alagille syndrome and the findings of liver biopsy.

Answer: The exclusion of other syndromes which clinical features similar to biliary atresia e.g., Alagille syndrome were performed by histopathology that revealed biliary hypoplasia, paucity of bile duct. (Page5 Line9)

4. In addition, the authors conducted a multivariate analysis, including the day of surgery and variant status. Since it has been shown that the anatomical pattern of biliary atresia also has a significant impact on prognosis, we believe that a multivariate analysis, including these factors, would provide important insights.

Answer: Together with the second question, we also included the anatomical factors into cox proportional hazards model calculation. The factors of age at Kasia operation and identified variant status were remained independently significantly increasing hazard risk. (Fig 5)

Attached with this mail, we submit the first revision of our manuscript. We are looking forward to hearing from the PLOS ONE.

Sincerely yours,

Surasak Sangkhathat, M.D., Ph.D.

Department of Surgery,

Faculty of Medicine, Prince of Songkla University,

Thailand 90110

---

## [Decision Letter · Decision Letter 1]

26 Oct 2022

A novel pathogenesis concept of biliary atresia approached by combined molecular strategies

PONE-D-22-12933R1

Dear Dr. Sangkhathat,

We’re pleased to inform you that your manuscript has been judged scientifically suitable for publication and will be formally accepted for publication once it meets all outstanding technical requirements.

Kind regards,

Gregory Tiao, M.D.

Academic Editor

PLOS ONE

Additional Editor Comments (optional):

Reviewers' comments:

Reviewer's Responses to Questions

**Comments to the Author**

1. If the authors have adequately addressed your comments raised in a previous round of review and you feel that this manuscript is now acceptable for publication, you may indicate that here to bypass the “Comments to the Author” section, enter your conflict of interest statement in the “Confidential to Editor” section, and submit your "Accept" recommendation.

Reviewer #1: All comments have been addressed

Reviewer #2: All comments have been addressed

2. Is the manuscript technically sound, and do the data support the conclusions?

Reviewer #1: Yes

Reviewer #2: Yes

3. Has the statistical analysis been performed appropriately and rigorously? 

Reviewer #1: Yes

Reviewer #2: Yes

4. Have the authors made all data underlying the findings in their manuscript fully available?

Reviewer #1: Yes

Reviewer #2: Yes

5. Is the manuscript presented in an intelligible fashion and written in standard English?

Reviewer #1: Yes

Reviewer #2: Yes

6. Review Comments to the Author

Reviewer #1: Well written paper suggesting novel perspective concept in the pathogenesis of BA by exploring genetic variations and biostatistical appraisal

I have no further suggestions

Reviewer #2: Thank you for responding to my comments.

The author replied properly.

The manuscript is well refined and I do not have further comments.

7. PLOS authors have the option to publish the peer review history of their article (what does this mean?). If published, this will include your full peer review and any attached files.

Reviewer #1: No

Reviewer #2: No
